# Time to Consider the “Exposome Hypothesis” in the Development of the Obesity Pandemic

**DOI:** 10.3390/nu14081597

**Published:** 2022-04-12

**Authors:** Victoria Catalán, Iciar Avilés-Olmos, Amaia Rodríguez, Sara Becerril, José Antonio Fernández-Formoso, Dimitrios Kiortsis, Piero Portincasa, Javier Gómez-Ambrosi, Gema Frühbeck

**Affiliations:** 1Metabolic Research Laboratory, Clínica Universidad de Navarra, 31008 Pamplona, Spain; vcatalan@unav.es (V.C.); arodmur@unav.es (A.R.); sbecman@unav.es (S.B.); 2CIBER Fisiopatología de la Obesidad y Nutrición (CIBEROBN), ISCIII, 31008 Pamplona, Spain; tono.fernandez@ciberisciii.es; 3Obesity and Adipobiology Group, Instituto de Investigación Sanitaria de Navarra (IdiSNA), 31008 Pamplona, Spain; 4Department of Neurology, Clínica Universidad de Navarra, 31008 Pamplona, Spain; iaviles@unav.es; 5Department of Nuclear Medicine, Medical School, University of Ioannina, 45110 Ioannina, Greece; dkiorts@uoi.gr; 6Clinica Medica “A. Murri”, Department of Biomedical Sciences and Human Oncology, University of Bari Medical School, 70124 Bari, Italy; piero.portincasa@uniba.it; 7Department of Endocrinology & Nutrition, Clínica Universidad de Navarra, 31008 Pamplona, Spain

**Keywords:** obesogens, “exposome”, environment, epigenetics, microbiota, antibiotics, viral infection, sleep, endocrine disruptors, brown adipose tissue, thermogenesis

## Abstract

The obesity epidemic shows no signs of abatement. Genetics and overnutrition together with a dramatic decline in physical activity are the alleged main causes for this pandemic. While they undoubtedly represent the main contributors to the obesity problem, they are not able to fully explain all cases and current trends. In this context, a body of knowledge related to exposure to as yet underappreciated obesogenic factors, which can be referred to as the “exposome”, merits detailed analysis. Contrarily to the genome, the “exposome” is subject to a great dynamism and variability, which unfolds throughout the individual’s lifetime. The development of precise ways of capturing the full exposure spectrum of a person is extraordinarily demanding. Data derived from epidemiological studies linking excess weight with elevated ambient temperatures, in utero, and intergenerational effects as well as epigenetics, microorganisms, microbiota, sleep curtailment, and endocrine disruptors, among others, suggests the possibility that they may work alone or synergistically as several alternative putative contributors to this global epidemic. This narrative review reports the available evidence on as yet underappreciated drivers of the obesity epidemic. Broadly based interventions are needed to better identify these drivers at the same time as stimulating reflection on the potential relevance of the “exposome” in the development and perpetuation of the obesity epidemic.

## 1. Introduction

If practitioners are asked about the current key public health challenges, in addition to the COVID-19 pandemic, many will mention obesity among the top priorities. The prevalence of obesity has tripled during the last decades, imposing an enormous burden not only on people’s health, but also on society at large with obesity increasing worldwide [1,2,3]. Risk factor exposure, relative risk, and imputable disease burden have been addressed in a comprehensive and standardized way by the Global Burden of Diseases, Injuries, and Risk Factors Study [4]. A rigorous analysis of the trends and specific levels of risk factor exposure together with a quantitative assessment of the plausible human health effects is of utmost importance. In this context, deep knowledge is required about when current efforts are being inadequate as opposed to when public health initiatives are showing fruitful effects. Identifying the ecological factors and external drivers of change that are currently tipping the balance may prove extraordinarily useful. This approach represents a biomedical challenge and public health need. Thus, it is worthwhile considering the conceptual basis to better understand alterations at the population level as well as their potential interaction with the surrounding with an innovative perspective on, as yet, underappreciated but conceivable factors. A search for original articles and reviews published between January 1990 and February 2022 focusing on causes and contributors was performed in PubMed and MEDLINE using the following search terms (or combination of terms): “obesity”, “epidemic or pandemic”, “comorbidity or comorbidities”, “outcomes”, “mortality”, “drivers”, “sedentarism”, “physical inactivity”, “environment or environmental”, “antibiotics”, “microbiota”, “genetics”, “epigenetics”, “viral infection”, “infectobesity”, “sleep”, “chronobiology”, “obesogens”, “endocrine disrupters”, “thermogenesis”, “urban planning”, “climate change” and “exposome”. Only English-language, full-text articles were included. Additional articles that were identified from the bibliographies of the retrieved articles were also used, as well as selected very recent references from March 2022. Articles in journals with explicit policies governing conflicts-of-interest, and stringent peer-review processes were favored. Data from larger replicated studies with longer periods of observation, when possible, were systematically chosen to be presented. More weight was given to randomized controlled trials, prospective case–control studies, meta-analyses and systematic reviews.

Up-to-date our thinking on the obesity epidemic has focused mainly on direct causes, such as genetic and behavioral determinants of energy intake and expenditure [5,6]. The combination of increased sedentarism and life expectancy have contributed to the obesity epidemic and its comorbidities with people exhibiting a poorer physical function [7,8,9]. Exercise produces extraordinarily complex physiological responses at the same time as inducing changes in cellular energy balance, leading to intensity-dependent activation of AMP-activated protein kinase (AMPK) in skeletal muscle [7,10], via effects on diverse intramuscular and hormonal factors adaptations to increased physical activity include amelioration of the cardiorespiratory fitness, as shown by an augmented maximal oxygen uptake together with an elevated muscle oxidative capacity promoted by an increased mitochondrial biogenesis and angiogenesis. Elicited signals include enhanced catecholamine signaling, sarcoplasmic calcium release, changes in mechanical stretch and force, metabolic alterations, disruptions to the redox state and acid–base balance, increased muscle temperature, and increased circulating adrenaline concentrations. These signals operate on transmembrane receptors, thereby activating downstream signaling pathways, or directly stimulate the release of exercise-responsive signaling molecules. Interestingly, exercise stimulates the secretion of metabolites, extracellular vesicles, and myokines that enable crosstalk with other organs, like adipose tissue, pancreas, liver, heart, gut, and brain as well as the vascular and immune systems.

When focusing on the time scale, two quite diverse influences can be distinguished that exert their effects on ingestive behavior, as well as on other aspects of energy homeostasis [11]. The evolutionary time frame, on the one hand, determines the selection of metabolic and behavioral traits embedded within a concrete genome. Famine, as a continuous peril to survival, has led to the selection of the so-called “thrifty genes”. Within a given environmental context, this thriftiness can be manifested at different levels, such as (i) the ‘energy-sparing’ metabolism to increase efficiency (metabolic), (ii) the proclivity to quick adipose tissue accretion (adipogenic), (iii) the capability to slow down or even switch off non-essential processes (physiologic), (iv) the propensity to hastily swallow available food (gluttony), (v) the proneness towards sedentarism to spare or conserve energy (sloth), and, finally, (vi) behavioural adaptations that can even result in selfish hoarding to warrant survival (Figure 1).

The life-course time frame, on the other hand, is responsible for determining the phenotype. The early embryo’s nutritional environment can exert major influences on its survival as well as on its short- and long-term physiological milieu. Thereafter, fetuses are still susceptible to nutritional intake, determined via the utero-placental unit and the maternal energetic supply. Through childhood, the adaptive plasticity is maintained and continues into adolescence and adulthood. Thereby, satiety and appetite, which encompass ingestive behavior, underlie a huge array of adaptations aimed first at survival. Thus, our “thrifty genes”, the “nutrition transition”, and the “technology-driven sedentariness” have been the main causes blamed, with regard to the obesity epidemic. However, recent mounting evidence obtained in diverse scientific settings is challenging this view. This narrative review reports the available evidence on the potential relevance of the “exposome” and the impact of yet underappreciated drivers of the obesity pandemic.

## 2. Emerging Evidence Working as Warning Signs

The past half-century has witnessed a particularly rapid increase in obesity, localized initially in high-income countries and urban settings, but also spreading, subsequently, to both low- and middle- income countries, as well as rural areas [3,12]. In this context, a conceptual framework may need to be put forward, focusing on more profound drivers embedded within society together with their interaction with biological, psychological, and socioeconomic processes.

### 2.1. Genetics

Rare, severe, early-onset monogenic obesity is often opposed to common or polygenic obesity as polarized and quite distinct entities. Studies for both forms of obesity, however, report shared genetic and biological underpinnings, thereby highlighting the pivotal role of the brain in body weight control [6]. New insights come from genome-wide association studies (GWAS) which are characterized by advanced sequencing technology in huge sample sizes. Moreover, cross-disciplinary post-GWAS approaches, combining novel analytical techniques and omics technologies, are opening new ways of understanding, and fostering the translation of genetic loci into meaningful biological pathways.

Genome-wide association scans for obesity-related traits have shown small size effects of the implicated genes that can be even reversed by physical activity [13,14,15,16,17]. Additionally, obesity appears to spread more through social than family ties [18], thereby further decreasing the relative relevance of genetics. On the contrary, the human genome is regulated via epigenetics whereby concrete ecological exposures bear risk for excess weight and associated comorbidities [19,20,21,22,23,24]. Given that survival of organisms is determined by the adequacy of nutrient intake to parallel energy expenditure, excess adiposity originally emerged as an advantageous developmental plasticity adaptation encompassing both intrauterine and intergenerational effects that bear maladaptive consequences in the current inappropriate scenario. While maternal nutrition and metabolism were well-established critical determinants of adult offspring health, adverse offspring outcomes are also reportedly associated with the father’s diet [25,26], thereby indicating non-genetic inheritance of paternal influence. In this sense, men with moderate obesity display distinct DNA methylation profiles as well as small non-coding RNA expression in sperm [27]. However, it is unknown to what extent epigenetic influences on gametes impact on the metabolic profile of the progeny. Moreover, lately, reproductive performance changes have taken place, including higher fertility among people with elevated fatness and increasing maternal age [28]. A noteworthy point is that the mother’s age influences excess weight risk via its impact on birth weight, whereby older women are at risk of delivering either larger or smaller babies as would be expected according to their gestational age, a circumstance that, in turn, augments the chances of originating adults with excess weight. In fact, the pregnant mother’s age and body mass index (BMI), as well as the father’s, together with the natal weight, the post-natal weight, and fat depot gain profiles reportedly exert an impact on the offspring’s life [29].

Assortative mating, i.e., the non-random mating of people as regards their phenotype and cultural factors, may have further contributed to the obesity epidemic [30]. The shift in the development of obesity earlier in time allows the univocal identification of partners with a specific phenotype concerning weight already in the late teens and early twenties [31]. Thus, the increase in excess weight evidenced recently in descendants may also relate to the impact of both simple and complex interactions on the non-random coupling of people based on BMI. People with high adiposity may go out with people with a similar phenotype and may be more comfortable as well as be attracted by persons with the same physical characteristics rather than by those with a normal weight. In addition, sharing the same sociocultural interests among people with similar BMI may also take place. Whilst matching of couples with excess body fat may accentuate the genetic susceptibility in the progeny, the underlying mechanism is still unclear [32]. Interestingly, married couples formed by people with elevated BMI already at school age have been shown to tend to increase alongside the excess weight pandemic, that, in turn, can elevate the progeny’s susceptibility to obesity [32].

### 2.2. Microbiome

The gut microbiome has also proven to be a key player in energy homeostasis [33,34], whereby specific gut microbial communities may be contemplated as another plausible factor for obesity development. Broad modifications in the gut microbiome have been evidenced in people with excess weight, which are reactive to changes in body weight [35,36,37,38]. Although a huge interindividual variability has been observed, in obesity, an overall reduction in microbial diversity, together with a particular decreased amount of *Bacteroidetes* at the same time as a consequent elevation of *Firmicutes*, have been reported. More precisely, observational obesity studies indicate less gut bacterial diversification with augmented levels of *Bacteroides fragilis*, *Fusobacterium, Lactobacillus reuteri,* and *Staphylococcus aureus*, at the same time as a lower representation of *Lactobacillus plantarum, Methanobrevibacter, Akkermansia muciniphila*, *Dysosmobacter welbionis*, and *Bifidobacterium animalis* in people living with obesity as compared to non-obese persons [38,39]. Mechanistically, the microbiome of people living with obesity has been associated with increasing energy-harvesting efficiency from the diet and alterations in gut permeability leading to metabolic endotoxemia, as well as changes in host gene expression that regulate inflammation, insulin resistance, fat storage, and fatty liver [40,41,42]. Latest findings indicate that microbiomes obtained from people with normal weight and obesity are different in how they interact with the host and its metabolism [43].

### 2.3. Infectobesity

Infection is getting more attention as a possible cause or inducing factor of obesity. The supporting findings come from both epidemiological data and the biological plausibility derived from the direct roles of some viral agents on reprogramming of the host’s metabolism towards adipogenesis. Over the past decades, evidence has been growing with regard to an increased incidence in children and adults living with obesity of both nosocomial and community-acquired infections, suggesting that specific infections may be involved in the development of obesity [44]. More recently, the COVID-19 syndemic has further shown how people living with obesity are more likely to become infected with the coronavirus SARS-CoV-2 and exhibit an elevated risk of hospitalization, complications, and mortality, in probable relation to an altered immune response to infection, a chronic low-grade inflammation, together with an increased cardiometabolic risk [45,46,47,48].

Viral infections, as well as by other microorganisms, have been put forward as a plausible explanation for the excess weight epidemic with the concept of “infectobesity” harbouring the possibility that some viruses and microbes may wield an etiological role in the development of obesity [49,50,51,52]. The specific impact of excess weight on the risk of infections and the immune response triggered by infections has been addressed in a small number of studies in the population with obesity [44,53,54]. It is noteworthy that obesity augments the susceptibility to infections via an impaired immune response [55]. In addition, excess weight can also affect the pharmacokinetics of antimicrobial drugs as well as the response to vaccines [56,57]. A direct role on the host’s metabolism reprogramming towards adipogenesis has been put forward as a causative or inducing factor of obesity. The existence of circulating antibodies against certain infectious agents (e.g., *Chlamydia pneumoniae* and adenovirus-36) has been associated with the suffering of excess weight [58,59]. Viral agents involved in the genesis of obesity can be classified into five main categories expanding from *Adenoviruses* and *Herpes* viruses to phages, slow viruses of transmissible spongiform encephalopathies, and other encephalitides, as well as hepatitides. Of all the viruses analyzed, adenovirus-36 (Ad-36) emerged as an appropriate candidate, according to clinical and modelling data [60]. Although mechanisms by which this adenovirus may prompt excess weight development need to be fully unraveled, it has been postulated that weight gain occurs via a direct adipogenic effect, whereby Ad-36 enters adipocytes modifying enzymatic and transcriptional factors leading to triacylglycerol accretion, increased oxidative stress, inflammation, and differentiation of preadipocytes into mature adipocytes [61,62]. A potential link between Ad-36 and obesity-related nonalcoholic fatty liver disease (NAFLD) development relies on leptin gene expression and insulin sensitivity reduction, glucose uptake increase, lipogenic and pro-inflammatory pathway activation in adipose tissue, and macrophage chemoattractant protein-1 elevation [63]. The possibility of the exchange of components of the microbiota, including the virome and virobiota, should not be discarded. In this context, the gut microbiota reportedly sustains intrinsic interferon signaling [63].

Of note, under persistent viral infections, the adaptation of the host’s metabolism and immunity may be jeopardized. In addition to fructose-rich diets, decreased insulin sensitivity, chronic systemic low-grade inflammation and mitochondrial alterations, and gastrointestinal microbiota are reportedly involved in the development and worsening of NAFLD [64,65,66]. Due to the affected hepatic metabolism, the secretion of organokines (adipokines, myokines, hepatokines, and osteokines, among others) can be altered [67]. Changes in the secretion pattern of hepatokines can indirectly or directly contribute to aggravating NAFLD. In particular, reciprocal alterations with a decrease in fibroblast growth factor (FGF) 19 and an increase in FGF21 concentrations have been reported in obesity [68,69]. Plausible organ-specific changes in the reactiveness to the FGFs are characteristic in excess weight with adipose and hepatic changes taking differing directions in β-Klotho expression.

### 2.4. Chronobiology

Energy balance conservation constitutes a dynamic process with circadian rhythmicity acting as a “timekeeper” playing a decisive role in systemic homeostasis [70]. Under physiological circumstances, clock-primed biological functions synchronize to anticipate daily demands to warrant survival. Light exposure, physical activity, and sleep patterns, as well as meal timing and composition are common factors involved in energy homeostasis. It is noteworthy that the disruption or desynchronization of these factors can favor the genesis of a wide number of non-communicable diseases (NCDs), among them obesity and its comorbidities [71]. Chronological features delineate the integration in time of prediction by clock genes and metabolic and bioenergetics reactions to nutrients, whereby molecular chronotypes might be further participating in the genesis of obesity.

The internal clock makes the organism ready for regular physiological functions, such as eating and sleeping, with alterations in clock priming causing disturbances in biological rhythms and metabolism [72]. The worldwide obesity prevalence increases and metabolism alterations concur with sleep debt together with an increase in shift work as well as night exposure to light [73,74,75,76]. Sleep curtailment, as well as alterations in the chronobiology, foster elevations in BMI and sabotage dietary efforts to diminish adiposity [77]. Lack of sleep was reportedly followed by augmented hunger, elevated circulating ghrelin concentrations, and decreased circulating leptin levels, when their energy intake was restricted, as opposed to when people were in positive energy balance. Moreover, reduced sleep reportedly impacts on numerous neuroendocrine signals coordinating substrate use such as the concentrations of catecholamines, thyroid, cortisol, and growth hormone. Sleep privation and sleep alterations relate to maladaptation of the hypothalamic–pituitary–adrenal axis, translating into increased production of glucocorticoids [78,79], which can compromise the immune system [80] and increase abdominal obesity in the long term [81]. It has been recently shown that people with excess weight curtailing their sleep regularly experienced a negative energy balance by extending their sleep duration in a real-life scenario [82]. A better knowledge of the interaction between circadian rhythm disturbance and energy homeostasis may help to explain the pathophysiological processes fundamental to weight gain, thereby paving the path towards identifying novel therapeutic approaches.

### 2.5. Endocrine Disrupters–Obesogens

The hypothesis relating to the evolutionary origination of well-being and sickness stems from decades ago [83,84]. Subsequently, diverse epidemiological studies evidenced the relation between maternal obesity while pregnant and the possibility of the progeny to develop certain chronic adult diseases or NCDs. Among the plausible underlying mechanisms, early developmental insulin resistance stands out. Additional factors include an increased placental nutrient transfer and fetal exposure to endocrine-disrupting chemicals (EDCs), which cross the placenta, exhibit diverse tissular bioaccumulation levels, and show gender-specific vulnerability, with male fetuses being more vulnerable than female ones [85,86]. In utero environment modifications may also underlie the transmittable epigenetic changes that can endure over various generations, thereby supporting the rationale for disease development later in life. Accumulating evidence shows that EDCs interfere with endocrine regulation and metabolism, leading to lifestyle-related cardiometabolic risk factors [87,88].

Interestingly, EDCs are widespread in the environment and our daily life, with exposure encompassing the air and foods, as well as habitual items as close as personal care products [89]. Whilst the effects of individual compounds have been extensively studied, the combination of chemicals needs to be analyzed in more detail to better understand the realistic landscape of exposure to EDCs. While a dose-response relationship has not been clearly established and may not always be predictable, accumulating evidence shows that already low exposures taking place in daily life may exert a notable impact on the individual’s susceptibility [90]. Moreover, in utero EDCs exposure exerts transgenerational effects reaching even the F4 generation [23]. EDCs impinge on pre- and postnatal growth, metabolism, body weight control, thyroid function, sexual development, puberty, and reproduction, among others. Though the exact mechanisms of how phenotypic features are transferred from an exposed organism to the progeny remain largely unknown strong evidence is mounting regarding a variety of epigenetic mechanisms including differential methylation of both DNA and histones, together with histone retention, non-coding RNAs expression and deposition, as well as chromatin organization and structure changes [23].

Obesity is positively associated with the exposure to EDCs [87,91]. The hypothesis of obesogens in the environment purports that pollutants of a chemical nature have the capacity to induce excess weight modifying metabolism and homeostatic set-points, affecting appetite regulation, altering lipid metabolism to stimulate adipocyte hypertrophy, and promoting adipogenic pathways aimed at fat cell hyperplasia, thereby predisposing, initiating or exacerbating weight gain [92,93]. Phthalates, per- and polyfluoroalkyl substances, polycyclic aromatic hydrocarbons, bisphenol A (BPA), heavy metals (cadmium, arsenic and mercury), and pesticides are well-known EDCs [94]. Important concepts regarding the potential impact of EDC include window and duration of exposure, role of combinations or mixtures, transgenerational effects, and epigenetic mechanisms. EDCs interrupt hormonal signaling, alter adipocyte differentiation, and interfere with metabolism, in particular during early developmental stages for several generations [94]. Various EDCs like BPA, diethylstilbestrol, phthalates and organotins, to mention a few, can interfere with signaling by targeting pathways of nuclear hormone receptors (glucocorticoid receptors, sex steroid, retinoid X receptor, and peroxisome proliferator-activated receptor γ) relevant to adipocyte proliferation and differentiation. At the adipocyte level, this is achieved by disrupting body weight homeostasis promoting long-term obesogenic changes with the epidemiological impact that can be multiplied when the interference takes place in moments of particular sensitivity like the fetal period and childhood. Thus, individuals exposed to obesogens may be preprogrammed towards an adipogenic fate worsened by socioeconomic circumstances favoring unhealthy diets as well as insufficient physical activity that promote poor diet and inadequate exercise and struggle lifelong to maintain a healthy weight. It is of note that BPA, polybrominated diphenyl ethers, phthalates, together with perfluoro products have been steadily increasing their levels in humans establishing a specific connection among adipogenic phenotypes with exposure and transcriptional network control [95].

The metabolism of xenobiotics is commonly viewed as a process of detoxification, but occasionally the metabolites of some compounds, which are usually inert or harmless, can become biologically active [96]. EDCs, in addition to stimulating adipogenesis and lipogenesis, can also repress lipolytic signaling, thereby inducing altered phenotypes [97]. Neurohormonal regulation of lipolytic rate classically underlies catecholamine-induced activation and insulin-stimulated suppression [98]. However, a large number of lipolytic mediators include mitogen-activated protein kinase, AMP-activated protein kinase, atrial natriuretic peptides, adipokines, and structural membrane proteins [99,100,101,102,103,104,105]. Among the latter ones, aquaglyceroporins (AQP3, AQP7, AQP9, and AQP10) represent a subfamily of aquaporins participating in glycerol movement across cell membranes. Due to their glycerol permeability, aquaglyceroporins are involved in energy balance. Glycerol influx and efflux control in metabolically relevant organs by aquaglyceroporins plays a pivotal role with the dysregulation of these glycerol channels being associated with metabolic diseases, such as obesity, insulin resistance, non-alcoholic fatty liver disease, and cardiac hypertrophy [106]. In fact, glycerol embodies a key metabolite as a substrate for de novo synthesis of triacylglycerols and glucose as well as an energy substrate for ATP production via mitochondrial oxidative phosphorylation. Noteworthy, the control of glycerol release by aquaglyceroporins in adipocytes plays a pivotal role in energy homeostasis reportedly associated with NCDs, such as insulin resistance, and obesity [107]. The potential interference of EDCs with a number of lipolytic factors deserves further analysis. Furthermore, EDCs also disrupt activity of brown and beige fat, the thermogenic adipose tissues [108].

Given the habitual exposure to multiple EDCs, the assessment of public health effects is complicated. In this respect, special care during pregnancy and childhood would be desirable. Sound knowledge about plausible mechanistic explanations on how specific exposures in a given environment translate into making individuals more susceptible to suffer some diseases like obesity [23]. Adequate determination of the surrounding toxicology together with its derived health risks might be achieved via advanced computational and prediction tools, and investigations of both systematic and integrative approaches further validating novel reliable metabolic biomarkers. Additionally, integration efforts aimed at mimicking the surrounding’s specific circumstances are needed in new studies pursuing the evaluation of the effects of EDCs.

### 2.6. Urban Planning

Interestingly, urban environment characteristics may also contain upstream drivers of obesity [109,110]. Nonetheless, consideration of the simultaneous combination of environmental factors is not normally addressed. When looking at the same time at 86 elements characterizing the urban “exposome” relating to BMI via geocoded exposures including individual home addresses, traffic noise, air pollution, built environment, and green-space, as well as neighborhood socio-demographic factors, relevant insight can be obtained. Exposure-obesity associations were identified after adjustment for individual socio-demographic characteristics. Associations of BMI with the mean neighborhood house cost, food facilities within a close reach, oxidation capacity of particulate elements, air pollution, low-income neighborhoods, and one-person households exhibited the strongest consistency [109]. BMIs were more elevated in low-income neighborhoods, in people with lower mean house cost, lower proportion of single-people households, and areas with lower numbers of healthy food facilities. The holistic analysis of the obesogens of the environment emphasizes the mounting information as regards the relevance of socioeconomics, urban planning, and air pollution as regards the neighborhood.

### 2.7. Climate Change

Global warming is a well-known public health challenge and bidirectional influences regarding adiposity and global warming have been established [111]. Since 1950, carbon emissions worldwide have increased at an exponential rate. Transport, construction, manufacturing, housing, forestry, and agriculture modifications, together with the world population increase in important obesity rates, can be considered as principal contributors to carbon emissions. With increasing atmospheric temperature, less adaptive thermogenesis can be expected in people with obesity who may simultaneously be less physically active, at the same time as increasing their carbon footprint. Thus, over the last centuries environmental influences like an increase in ambient temperature in relation to climate change and global warming together with transportation, temperature insulation of both edifices, and individuals have decreased the necessity of people to generate energy by inducing thermogenesis. Therefore, it is important to consider the environmental impact of the rising obesity rates to learn more about how to tackle the excess weight pandemic, at the same time as how to minimize energy consumption, food waste, greenhouse gas emissions, in general, and carbon footprint, in particular. Of note, the Mediterranean diet, which is characterized by low in meat intake, reportedly reduces by 72% greenhouse gas emissions, by 58% land use, and by 52% energy consumption [111].

In this context, it is important to consider that human fat consists mainly of white adipose tissue (WAT) and brown adipose tissue (BAT) [112]. Whereas WAT stores energy surplus and releases it according to the needs of the organism, BAT converts it to heat playing a role in body temperature control [113,114,115,116,117]. Patches of brown-like adipocytes that appear in WAT constitute beige fat [118]. Like BAT, beige fat also represents a further thermogenic adipose depot with increased levels of thermogenic genes and respiration rates. Interestingly, beige adipocytes, also termed brite (derived from the contraction of “brown-in-white”) cells, resemble white adipocytes in the basal state, but are rich in mitochondria and release heat when activated in response to thermogenic stimuli [119]. Thus, beige or brite adipocytes exhibit a distinct gene expression pattern to that of brown or white fat cells. The worldwide temperature increase might be also playing a role in the obesity epidemic via a concomitant reduction in BAT activity [120]. BAT as well as beige fat have been greatly underestimated in adults. For many years the BAT contribution in adults to energy expenditure both in terms of amount and effectiveness was presumed to be trivial due to the presence of only marginal brown fat depots [114]. BAT and beige fat express uncoupling protein 1 (UCP1), which rapidly generates heat when activated. UCP1 is stimulated by cold-exposure and diet leading to increased activity of the sympathetic nervous system as well as oxidation of huge quantities of glucose and lipids. The identification of functional BAT in adult humans that can be stimulated by cold exposure has changed our understanding of cellular bioenergetics, especially with regard to adaptive thermogenesis in humans [113,114,115,116,117].

Whilst BAT research has mainly addressed its participation in non-shivering thermogenesis, the identification of its highly dynamic secretory capacity has revealed its endocrine and paracrine function via the release of “batokines” [121]. These plentiful BAT-derived molecules impinge on the physiology of diverse cell types and multiple organ systems like adipose tissue, skeletal muscle, liver, and cardiovascular system, among others [122]. Interestingly, the variety of signaling molecules encompassed by batokines extends from peptides and lipids to metabolites and microRNAs [123]. Further research in humans aimed at delineating the role of batokines beyond the BAT-mediated energy expenditure is required. Among the endocrine batokines peptide factors like adiponectin, FGF21, interleukin-6, neuregulin-4, myostatin, and phospholipid transfer protein, as well as some microRNAs like miR-92a and miR-99b, stand out, whereas the lipids include lipokines, bioactive compounds, derived from adipose tissue, that regulate diverse molecular signaling pathways. Recently, an oxylipin, 12,13-dihydroxy-9Z-octadecenoic acid (12,13-diHOME), has attracted interest. The elevation in serum 12,13-diHOME has been associated with improved metabolic health with the action of this molecule appears to be mediated by brown adipose tissue (BAT). Its circulating concentrations are negatively correlated with BMI and insulin sensitivity. Exposure to cold and physical exercise result in an increase in circulating levels of 12,13-diHOME, which promotes browning of WAT and stimulates fatty acid absorption by BAT via stimulating the translocation of the fatty acid transporters CD36 and FATP1 to the cell membrane [124,125]. Moreover, the existence of other as yet unidentified factors involved in energy balance regulation should not be discarded [126,127].

### 2.8. Plurality of Obesity Epidemics

A noteworthy, elegant cross-species analysis has clearly shown a plurality of epidemics of excess weight among domestic mammals, even without the presence of the elements characteristically conceived as the main predetermining factors of the obesity epidemic via their impact on lifestyle habits like diet and physical activity [128]. These findings indicate that excess weight genesis over the last decades depends on the confluence of additional yet underappreciated environmental influences.

## 3. The “Exposome” as a Plausible Underlying Mechanism of Action

The word “exposome” stands for the assessment over the whole life of a person of all the exposures and its relationship to disease. This concept has been fostered by the success in mapping the human genome [129,130]. Of note, the exposure of a person starts at conception and in utero, continuing over childhood and adolescence (Figure 2). Job-related insults as well as influences from leisure time and the environment further accumulate during adulthood progressing up to senescence. Many single nucleotide polymorphisms (SNPs) are genetic variants of low penetrance involved in the control of food intake, body weight, and lipid metabolism, among others.

Despite their low penetrance, the SNPs’ high prevalence implies a potential substantial contribution to the disease burden at the population level. This means that in a concrete exposure scenario the majority of SNPs, although being of low penetrance, will emerge because of strong environmental influences. While exposures of the surrounding exhibit an exceedingly relevant protagonism in the development of NCDs, a clear association is not easy to unravel. The “exposome” will be best deciphered by obtaining deeper knowledge on how dietary and lifestyle exposures interplay with the individual’s unique genetic, epigenetic, and physiologic characteristics translate into disease. In this scenario, the “exposome” can be contemplated from a conventional point of view, in which insults are randomly distributed along the whole lifecycle, or with the lens of the critical window exposure, in which insults are non-randomly allocated to specific time-periods during life [131]. Improvement in disease etiology identification at the population level will come from complementing the emphasis on genotyping by a detailed analysis of the plentiful environmental exposures [132,133], with its accurate assessment remaining a formidable and pending demand in obesity assessment. Moreover, the development of methods that accurately capture both the external environment as well as the internal chemical background of the individual are urgently needed (Figure 3). In order to complement the “genome” with its matching “exposome” the same precision for an individual’s environmental exposure as we have for the subject’s genome should be pursued.

### Need for an Integral Consideration of the Collective Impact of Simultaneously Acting Drivers

Contrarily to the genome, the “exposome” is subject to a great dynamism and variability, which unfolds throughout the individual’s lifetime. The development of precise ways of determination that capture the full exposure spectrum of a person is extraordinarily demanding. These considerations are particularly relevant for children and adolescents with obesity, given that the increased exposure is expected to translate into larger adverse effects than weight gain only during adulthood [134,135]. Furthermore, the concept of epigenetics comprises the study of changes in the organism caused by alterations in gene expression rather than modifications of the genetic code itself [136,137]. Interestingly, epigenetic marks can be affected by air pollution, organic pollutants, exposure to benzene, metals, and electromagnetic radiation. Other potential environmental stressors capable of changing the epigenetic landscape include chemical and xenobiotic compounds present in the atmosphere or water.

Moreover, while responses to certain specific exposures are invariable, to other external insults responses may change (“resposome”), with disparity depending on genome and epigenome changes (Figure 4). While some alterations reveal chronicity in exposure, certain cases reveal a latent response, based on “priming” for a late pathogenesis via epigenetic changes.

Analysis of the current human “exposome” emphasizes the challenges represented by the concepts of lifelong exposure and the need to compute all environmental factors in order to obtain the whole real life exposomic scenario [131]. To overcome these limitations and establish the relation between human health and the “exposome” focusing on critical-window periods can be combined with data- and hypothesis-driven exposomics. Moreover, analysis of high-throughput and multidimensional data of both internal and external exposure factors are welcome [131]. Useful tools to analyze the “exposome” and foster exposomics should comprise different steps, i.e., (i) the development of biomarkers capturing exposure effect, susceptibility to exposure, and disease progression; (ii) the application of advances that integrate systems biology with environmental big data; and (iii) exploratory data mining to analyze the relationships between exposure effects, and other factors that ultimately lead to obesity development and thereby provide potential mechanistic information (Figure 5). Artificial intelligence will broadly reshape medicine, thereby improving the experiences of both patients and clinicians. In fact, artificial intelligence is already being applied in an ever-increasing number of medical fields moving from what might have been considered speculation years ago to reality right now. Progress in data analysis, including image deconvolutions, non-image data sources, unconventional problem formulations, sophisticated algorithms, and human–artificial intelligence collaborations, will reduce the gap between research and clinical practice. While these challenges are being addressed, artificial intelligence will develop exponentially, making healthcare more accessible, efficient, and accurate for patients worldwide [138].

In order to be particularly helpful, “exposome” assessment should combine GWAS together with epigenome-wide association trials and detailed metabolic-endocrinological phenotyping of the individuals. Moreover, these combined analyses should be applied at multiple time-points to establish the potential interaction effect. The large amount of data on exposures provided by these projects hinders the interpretation of their relationship with health outcomes and omics. In this regard, similar or parallel databases to genetics (OMIN, dbSNP, or TCGA) may be developed for exposomics. Together with handling and archiving large data volumes, the lack of standard nomenclature, the quality of output from each analytical platform, or the heterogeneity of data constitute important issues to be resolved. Given the important public health problem posed by the rise in NCDs like obesity, the presented proposal of integration of elements that constitute the “exposome” will strengthen the better comprehension of the intricate underlying mechanisms, thereby opening pathways to innovative preventive and therapeutic strategies.

## 4. Conclusions

The more simplistic energy balance model of obesity has been surpassed by epidemiological, biological, psychological, and socioeconomic evidence. Far-reaching holistic modelling of obesity is required in order to establish effective interventions aimed at its efficient treatment and better prevention. The origins of excess weight are rooted in an extremely complicated biological network, set within a similarly intricate societal and environmental organization (Figure 6), which needs to be carefully considered.

Analysis of alternative and less researched etiologies is needed. The gut microbiome, circadian rhythms, and infectobesity, to mention only a few, constitute other candidate alternate etiologies. More multidisciplinary, translational research must analyze the intricacies of such alternate etiologies, as well as develop unprecedented stratagems for fending off a multifactorial and plurietiological pathology via, for example, prioritization of root cause interrogation and group risk assessment. Knowledge gaps persist in this relevant area whereby a comprehensive, leveraged patient-centered research would be welcome. Due to the struggle in the coming years to override the key factors steering the present excess weight epidemic, an inclusive, detailed, pro-active, durable program and fresh perspectives to unravel the whole panoply of causative factors is needed to outline a feasible counter reply to manage the defiance imposed by the pandemic. A comprehensive understanding of the causative factors of obesity might provide more effective management approaches.

## Figures and Tables

**Figure 1 nutrients-14-01597-f001:**
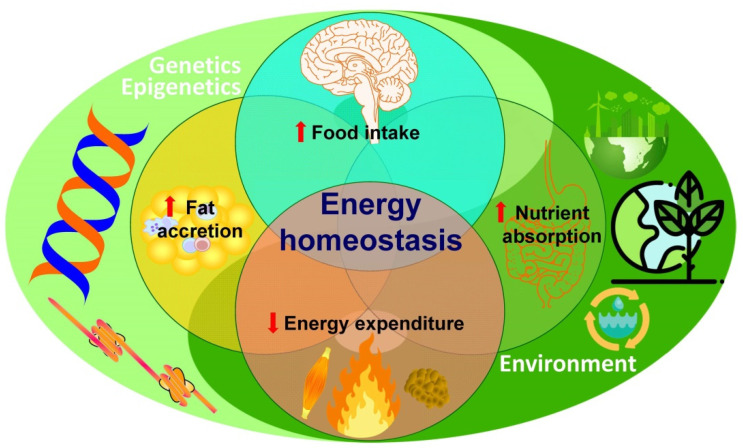
**Schematic diagram of the factors involved in energy homeostasis.** The classical Venn diagram shows how in obesity the intersection between increased food intake, nutrient absorption, and fat accumulation, together with decreased energy expenditure, the main factors determining energy homeostasis, are simultaneously under the broader influence of the environment as well as genetics and epigenetics.

**Figure 2 nutrients-14-01597-f002:**
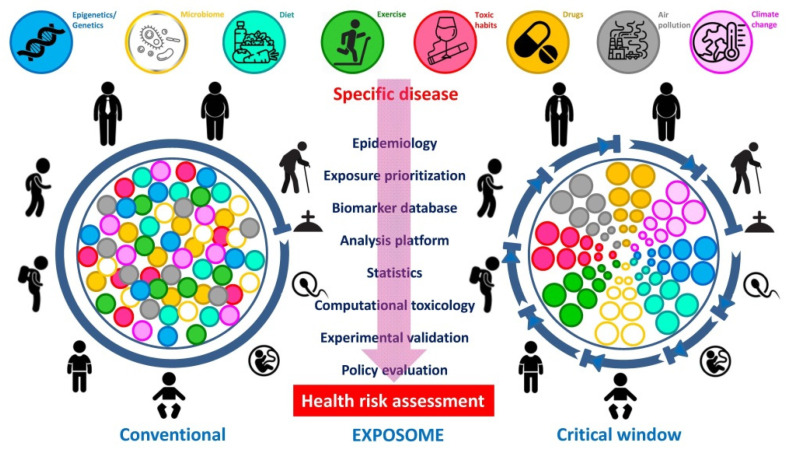
**Conventional versus critical window “exposome” views in assessment of health risk in humans.** The comprehensive portrait of an individual’s “exposome” evolves throughout the lifetime with the possibility of prioritized exposure factors, during specific time-points or critical windows as opposed to a standard, random or linear exposure. The diagram emphasizes the relevance of measurements at different time-points (modified from Fang et al. [131]).

**Figure 3 nutrients-14-01597-f003:**
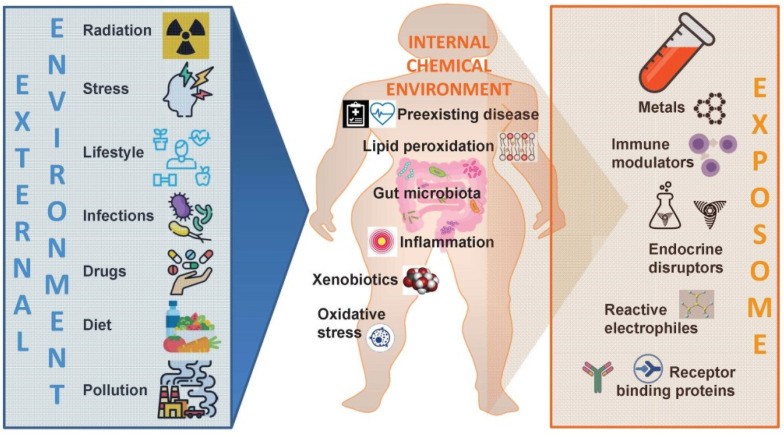
**Characterization of the “exposome”.** The “exposome” of a given person represents the combined exposures from all external sources that reach the internal chemical environment. Specific biomarkers or potential signatures of the “exposome” might be detected in the bloodstream.

**Figure 4 nutrients-14-01597-f004:**
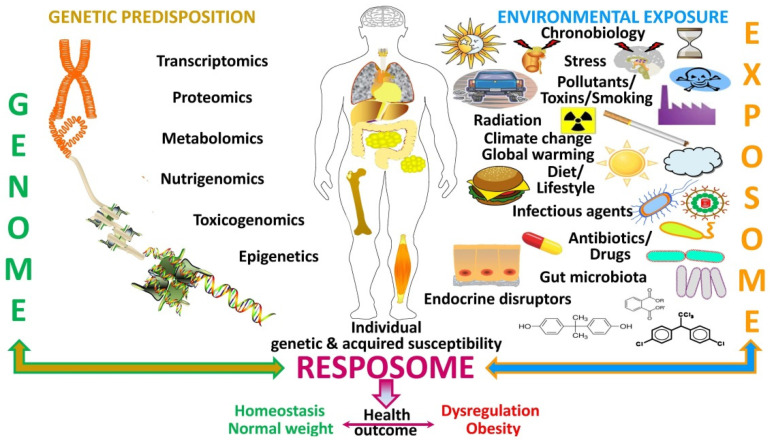
**Factors influencing the resposome.** Schematic diagram on how the genetic predisposition (genome) interacts with the environmental exposure (“exposome”) to influence an individual’s genetic and acquired susceptibility shaping its responses (resposome) that yield the ultimate health outcome as regards body weight control.

**Figure 5 nutrients-14-01597-f005:**
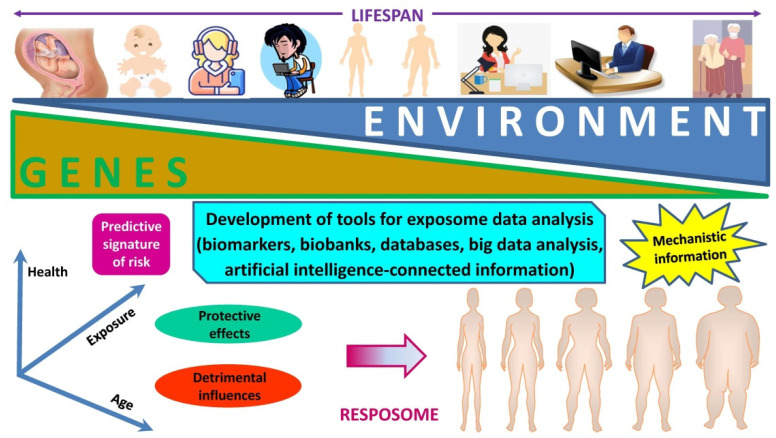
**Evolution of the individual’s genetic and environmental framework across the lifespan.** Over a lifetime, genetic and environmental influences may change reciprocally with acute and chronic exposures translating into a specific information with predictive interest as well as effective biomarkers that may provide mechanistic insight of pragmatic application.

**Figure 6 nutrients-14-01597-f006:**
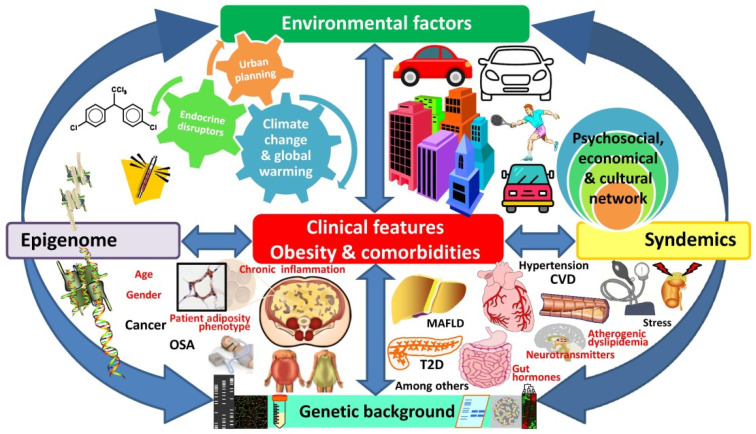
Multidimensional view of the complex interaction of the main drivers involved in excess weight development and obesity-associated comorbidities. OSA, obstructive sleep apnea; MAFLD, metabolic-associated fatty liver disease; T2D, type 2 diabetes; CVD, cardiovascular diseases.

## Data Availability

Not applicable.

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
