# Peer review of "Time to Consider the “Exposome Hypothesis” in the Development of the Obesity Pandemic"

_nutrients, 2022, doi:10.3390/nu14081597_

Round 1
Reviewer 1 Report
This is a welcome summary of the obesity epidemic. More specifically, the review is trying to give a perspective on the underappreciated obesogenic factors, which authors propone to refer to as the “exposome”. The authors after an accurate exploration of the literature provide detailed information on this topic. Although of interest the manuscript lacks an important aspect that should be inserted concerning the unphysical activity of obese people and the role of metabolism from a biochemistry point of view.
Author Response
We would like to thank the Reviewer for the positive comments.
Although several recent reviews (see below) have focused on the relation between obesity and physical inactivity, following the Reviewer’s indication, we have also incorporated this aspect to the revised manuscript.
Gielen S, Schuler G, Adams V. Cardiovascular effects of exercise training: molecular mechanisms. Circulation 2010;122:1221-38.
Neufer PD et al. Understanding the cellular and molecular mechanisms of physical activity-induced health benefits.Cell Metab 2015;22:4-11.
Hojman P, Gehl J, Christensen JF, Pedersen BK. Molecular mechanisms linking exercise to cancer prevention and treatment. Cell Metab 2018;27:10-21.
Calderón-Larrañaga A, Vetrano DL, Ferrucci L, Mercer SW, Marengoni A, Onder G, Eriksdotter M, Fratiglioni L. Multimorbidity and functional impairment-bidirectional interplay, synergistic effects and common pathways. J Intern Med. 2019 Mar;285(3):255-271.
Thyfault JP, Rector RS. Exercise combats hepatic steatosis: Potential mechanisms and clinical implications. Diabetes 2020;69:517-24.
Moreira JBN, Wohlwend M, Wisloff U. Exercise and cardiac health: physiological and molecular insights. Nat Metab 2020;2:829-39.
McGee SL, Hargreaves M. Exercise adaptations: molecular mechanisms and potential targets for therapeutic benefit. Nat Rev Endocrinol 2020;16:495-505.
Dixon BN, Ugwoaba UA, Brockmann AN, Ross KM. Associations between the built environment and dietary intake, physical activity, and obesity: A scoping review of reviews. Obes Rev 2021;22:e13171.
Woessner MN, Tacey A, Levinger-Limor A, Parker AG, Levinger P, Levinger I. The Evolution of Technology and Physical Inactivity: The Good, the Bad, and the Way Forward. Front Public Health. 2021 May 28;9:655491
Diniz TA, Antunes BM, Little JP, Lira FS, Rosa-Neto JC. Exercise Training Protocols to Improve Obesity, Glucose Homeostasis, and Subclinical Inflammation. Methods Mol Biol. 2022;2343:119-145.
Reviewer 2 Report
While the importance of the "Exposome Hypothesis" is discussed from multiple perspectives, there is little explanation how to select previous studies in obesity, and the review methodology should be reconsidered.
Author Response
We would like to thank the Reviewer for the positive comments.
Following his/her pertinent indication, we have included the search strategy followed for the selection of the studies on obesity.

Reviewer 3 Report
The manuscript by Catalán et al., entitled “Time to Consider the “Exposome Hypothesis” in the Development of the Obesity Pandemic”, reference number Nutrients - 1642339 is an interesting manuscript, well thought out, with fine scientific writing, and technically sound. The topic is highly relevant and updated. In fact, the obesity epidemic, with its prohibitive health care and social costs, and the poor success record of available interventions, either behavioral or pharmacological, has prompted the search for novel therapeutic strategies, efficient treatment and better prevention. In this review manuscript, the authors present several lines of evidence acting in an extremely intricate biological network underlying the exposure to underappreciated obesogenic factors, collectively called as “exposome” as responsible also for excess weight. While genetics and overnutrition along with a sedentary lifestyle remain causative factors for obesity onset and development, the “exposome hypothesis” is subject to great dynamics and variability throughout an individual’s lifetime. Despite the novelty and interest, I think the manuscript is a little bit long, dense and sometimes exhausting for the reader. For this reason, I suggest cutting the manuscript by 25% and simplify some descriptions. In what concerns the bibliography, the literature is well cited, fairly covered with recent bibliographic references, including the years of 2019, 2020, 2021 and even 2022. I also would like to congratulate the authors for Figures design which are original and informative. In view of the stated above, the manuscript requires some revisions.
Apart from these general comments, along I was reading the manuscript some major and minor considerations came up. And here they are, point by point:
Major and minor comments:
- In the Abstract section, please replace “culprits” by another word (page 1, line 21).
- “Sleep curtailment” or sleep privation? (Abstract section, page 1, line 29).
- Still in the Abstract, “drivers” or triggers? (page 1, lines 32 and 33).
- Reviewer’s suggestion: I think it would be interesting to include a glossary in this manuscript, providing the definition of many concepts herein applied, such as obesogens, exposome, resposome, thrifty genes, NCDs, EDCs, batokines, and so on.
- The first question that came into my mind when I was reading this manuscript was: how does the concept of obesogenic environment/obesogenic factors relate to the “exposome hypothesis”? (see page 7, lines 295-298).
- In Figure 1 about the classical Venn diagram, please provide some bibliographic references (page 2, line 71).
- Also for Figure 1, please replace “Absorption” by “Nutrient absorption” (page 2).
- The “exposome” must be written in quotation marks throughout the manuscript (see for example, Figure 2 and Figure 3 legends).
- Regarding the methodology, the authors should include a sentence like this: The research publications reviewed in the present study were all obtained from Web of Science (Clarivate Analytics, Phyladelphia, PA, USA) from X to Y of the following years Z.
- What about the concept of “metabolically healthy obesity”?
- Define “thrifty genes” (page 3, line 83).
- Please consider the sentence: “In this context, a conceptual framework may need to be put forward, focusing on more profound drivers embedded within society together with their interaction with biological, psychological, socioeconomic, and geopolitical processes” (page 3, lines 91-94). Under this context, please explain the role of geopolitical processes.
- Overall, in my opinion the manuscript lacks of certainty and it is a little bit speculative. Avoid speculations through the entire manuscript, like: “ We can speculate…” (page 4, line 135)
- Please, replace “triglyceride” by “triacylglycerols” (page 5, line 194).
- From the authors’ point of view: how does the theory of “adipose tissue expandability” fit in this manuscript? How does the theory of “adipose tissue expandibility” relate to the “exposome hypothesis” in the development of obesity? (see, Virtue S, Vidal-Puig A. It's not how fat you are, it's what you do with it that counts. PLoS Biol. 2008; 6 (9): e237; Vidal-Puig A. Adipose tissue expandability, lipotoxicity and the metabolic syndrome. Endocrinol. Nutr. 2013; 60 (Suppl 1): 39-43; Gray SL, Vidal-Puig A. Adipose tissue expandability in the maintenance of metabolic homeostasis. Nutr. Rev. 2007; 65 (6): S7-S12).
- Please, define NCDs, non-communicable diseases (page 5, line 222).
- Although I think the manuscript is a bit long and should be shorten by 25%, I would like to see the aquaglyceroporins theme further exploited (page 7, lines 310-313). Moreover, highly valuable research teams across Europe working with aquaglyceroporins and obesity deserve to be cited.
- Please, provide more information on the lipokine, 12,13-dihydroxy-9Z-octadecenoic acid (page 8, line 392).
- Please replace “tsunami” by another word (page 9, line 401).
- Congratulations on Figures design, in particular Figure 2 (page 9).
- Please integrate the concept of “epigenetics” before the sentence in lines 448-451 (page 10).
- Another suggestion from this reviewer is to include the word lifespan at the top of Figure 5 (page 11).
- Define “critical-window period” (page 11, line 463).
- Consider the legend of Figure 5: again, are we being too speculative? I mean: can the potential signature (line 475, page 11) be translated into a scale? An analytical number? Do you think it will be possible to create/develop a scale in the future?
- Still regarding Figure 5, how can artificial intelligence-connected information contribute to the development of tools for “exposome” data analysis? (page 11).
- Please, reformulate the sentence: “The purported integration proposal will strengthen the better comprehension of the intricate mechanisms underlying the rise of NCDs like obesity, which pose such an important public health problem, and probably would lead to innovative preventive and therapeutic strategies” (page 11, lines 486-489).
- Please, provide a comparison between NAFLD, non-alcoholic fatty liver disease and MAFLD, metabolic-associated fatty liver disease supported by adequate bibliographic references (page 12, line 499).
- In my opinion, the Conclusions are very well presented (page 12, lines 490-513).
Author Response
We would like to thank the Reviewer for the encouraging comments and detailed editing, which have served to improve the manuscript.
Apart from these general comments, along I was reading the manuscript some major and minor considerations came up. And here they are, point by point:
Major and minor comments:
- In the Abstract section, please replace “culprits” by another word (page 1, line 21).
The word “culprits” has been replaced by “causes”.
- “Sleep curtailment” or sleep privation? (Abstract section, page 1, line 29).
Sleep curtailment is a frequently used term in this context (please see some examples below) used to define a shortened sleep relative to average sleep duration, as opposed to privation, which refers to the situation or condition of suffering from a lack of sleep.
Lucassen EA, Rother KI, Cizza G. Interacting epidemics? Sleep curtailment, insulin resistance, and obesity. Ann N Y Acad Sci. 2012 Aug;1264(1):110-34.
Zimberg IZ, Dâmaso A, Del Re M, Carneiro AM, de Sá Souza H, de Lira FS, Tufik S, de Mello MT. Short sleep duration and obesity: mechanisms and future perspectives. Cell Biochem Funct. 2012 Aug;30(6):524-9.
Porkka-Heiskanen T, Zitting KM, Wigren HK. Sleep, its regulation and possible mechanisms of sleep disturbances. Acta Physiol (Oxf). 2013 Aug;208(4):311-28.
Copinschi G, Caufriez A. Sleep and hormonal changes in aging. Endocrinol Metab Clin North Am. 2013 Jun;42(2):371-89.
Nedeltcheva AV, Scheer FA. Metabolic effects of sleep disruption, links to obesity and diabetes. Curr Opin Endocrinol Diabetes Obes. 2014 Aug;21(4):293-8.
Van Someren EJ, Cirelli C, Dijk DJ, Van Cauter E, Schwartz S, Chee MW. Disrupted Sleep: From Molecules to Cognition. J Neurosci. 2015 Oct 14;35(41):13889-95.
St-Onge MP, Grandner MA, Brown D, Conroy MB, Jean-Louis G, Coons M, Bhatt DL; American Heart Association Obesity, Behavior Change, Diabetes, and Nutrition Committees of the Council on Lifestyle and Cardiometabolic Health; Council on Cardiovascular Disease in the Young; Council on Clinical Cardiology; and Stroke Council. Sleep Duration and Quality: Impact on Lifestyle Behaviors and Cardiometabolic Health: A Scientific Statement From the American Heart Association. Circulation. 2016 Nov 1;134(18):e367-e386.
Blake-Lamb TL, Perkins ME, Taveras EM. Risk Factors for Childhood Obesity in the First 1,000 Days: A Systematic Review. Am J Prev Med. 2016 Jun;50(6):761-779.
Viot-Blanc V. Sleep duration and metabolism. Rev Mal Respir. 2015 Dec;32(10):1047-58.
Mosavat M, Mirsanjari M, Arabiat D, Smyth A, Whitehead L. The Role of Sleep Curtailment on Leptin Levels in Obesity and Diabetes Mellitus. Obes Facts. 2021;14(2):214-221.
Pizza F, Filardi M, Moresco M, Antelmi E, Vandi S, Neccia G, Mazzoni A, Plazzi G. Excessive daytime sleepiness in narcolepsy and central nervous system hypersomnias. Sleep Breath. 2020 Jun;24(2):605-614.
- Still in the Abstract, “drivers” or triggers? (page 1, lines 32 and 33).
As in the previous point, the term “drivers” is more frequently used in the context of obesity (please, see some publications below).
Pavlovska I, Polcrova A, Mechanick JI, Brož J, Infante-Garcia MM, Nieto-Martínez R, Maranhao Neto GA, Kunzova S, Skladana M, Novotny JS, Pikhart H, Urbanová J, Stokin GB, Medina-Inojosa JR, Vysoky R, González-Rivas JP. Dysglycemia and Abnormal Adiposity Drivers of Cardiometabolic-Based Chronic Disease in the Czech Population: Biological, Behavioral, and Cultural/Social Determinants of Health. Nutrients. 2021 Jul 8;13(7):2338.
Jian C, Carpén N, Helve O, de Vos WM, Korpela K, Salonen A. Early-life gut microbiota and its connection to metabolic health in children: Perspective on ecological drivers and need for quantitative approach. EBioMedicine. 2021 Jul;69:103475.
Laso A, Gutiérrez-Larrañaga M, Alonso-Peña M, Medina JM, Iruzubieta P, Arias-Loste MT, López-Hoyos M, Crespo J. Pathophysiological Mechanisms in Non-Alcoholic Fatty Liver Disease: From Drivers to Targets. Biomedicines. 2021 Dec 26;10(1):46. doi: 10.3390/biomedicines10010046.
Bond ST, Calkin AC, Drew BG. Sex differences in white adipose tissue expansion: emerging molecular mechanisms. Clin Sci (Lond). 2021 Dec 22;135(24):2691-2708.
Adom T, De Villiers A, Puoane T, Kengne AP. A Scoping Review of Policies Related to the Prevention and Control of Overweight and Obesity in Africa. Nutrients. 2021 Nov 11;13(11):4028.
Chavez-Ugalde Y, Jago R, Toumpakari Z, Egan M, Cummins S, White M, Hulls P, De Vocht F. Conceptualizing the commercial determinants of dietary behaviors associated with obesity: A systematic review using principles from critical interpretative synthesis. Obes Sci Pract. 2021 Apr 5;7(4):473-486.
McCarthy D, Berg A. Weight Loss Strategies and the Risk of Skeletal Muscle Mass Loss. Nutrients. 2021 Jul 20;13(7):2473.
- Reviewer’s suggestion: I think it would be interesting to include a glossary in this manuscript, providing the definition of many concepts herein applied, such as obesogens, exposome, resposome, thrifty genes, NCDs, EDCs, batokines, and so on.
Taking into account the suggestion of the Reviewer a glossary of the main concepts dealt with in the manuscript has been included in the revised version of the manuscript.
- The first question that came into my mind when I was reading this manuscript was: how does the concept of obesogenic environment/obesogenic factors relate to the “exposome hypothesis”? (see page 7, lines 295-298)..
Obviously, the obesogenic environment and factors, which encompass lifestyle changes including fast-food, high-fat & energy-dense diets, sedentary behaviour and urban planning, conform part of the exposome. Thus, the obesogenic environment includes usually the classical factors related to the development of obesity. However, as explained in the manuscript, the exposome hypothesis opens up to other as yet underappreciated factors placing particular emphasis on the concurrence of multiple of these less considered elements during the last years, which happen to operate over the whole lifespan.
- In Figure 1 about the classical Venn diagram, please provide some bibliographic references (page 2, line 71).
We do not understand the suggestion of providing some bibliographic references in Figure 1 since this is an original figure made by the authors.
- Also for Figure 1, please replace “Absorption” by “Nutrient absorption” (page 2).
As indicated, “Absorption” has been replaced by “Nutrient Absorption” in Figure 1.
- The “exposome” must be written in quotation marks throughout the manuscript (see for example, Figure 2 and Figure 3 legends
As indicated, “exposome” has been written in quotation marks throughout the manuscript.
- Regarding the methodology, the authors should include a sentence like this: The research publications reviewed in the present study were all obtained from Web of Science (Clarivate Analytics, Phyladelphia, PA, USA) from X to Y of the following years Z.
Following the reviewer’s indication, we have included the search strategy followed for the selection of the studies on obesity.
- What about the concept of “metabolically healthy obesity”?
At this point, we could only speculate on how “metabolically healthy obesity” relates to the “exposome hypothesis”. Since the Reviewer has adviced to eliminate speculation as much as possible, we will refrain from providing a potential explanation. Nonetheless, it has to be mentioned that the concept of metabolically healthy obesity is far from having a univocal definition, thereby providing slippery ground to make robust assumptions.
- Define “thrifty genes” (page 3, line 83)
See the Glossary.
- Please consider the sentence: “In this context, a conceptual framework may need to be put forward, focusing on more profound drivers embedded within society together with their interaction with biological, psychological, socioeconomic, and geopolitical processes” (page 3, lines 91-94). Under this context, please explain the role of geopolitical processes.
Following the reviewer’s comment, the term “geopolitical” has been deleted.
- Overall, in my opinion the manuscript lacks of certainty and it is a little bit speculative. Avoid speculations through the entire manuscript, like: “ We can speculate…” (page 4, line 135)
Speculations have been omitted.
- Please, replace “triglyceride” by “triacylglycerols” (page 5, line 194).
Done.
- From the authors’ point of view: how does the theory of “adipose tissue expandability” fit in this manuscript? How does the theory of “adipose tissue expandibility” relate to the “exposome hypothesis” in the development of obesity? (see, Virtue S, Vidal-Puig A. It's not how fat you are, it's what you do with it that counts. PLoS Biol. 2008; 6 (9): e237; Vidal-Puig A. Adipose tissue expandability, lipotoxicity and the metabolic syndrome. Endocrinol. Nutr. 2013; 60 (Suppl 1): 39-43; Gray SL, Vidal-Puig A. Adipose tissue expandability in the maintenance of metabolic homeostasis. Nutr. Rev. 2007; 65 (6): S7-S12).
We could only speculate on how the theory of “adipose tissue expandibility” would fit with the “exposome hypothesis” in the development of obesity. Since the Reviewer has adviced to eliminate speculation as much as possible we will refrain from providing a potential explanation.
- Please, define NCDs, non-communicable diseases (page 5, line 222).
See the Glossary.
- Although I think the manuscript is a bit long and should be shorten by 25%, I would like to see the aquaglyceroporins theme further exploited (page 7, lines 310-313). Moreover, highly valuable research teams across Europe working with aquaglyceroporins and obesity deserve to be cited.
We have tried to shorten the manuscript wherever possible. However, following the suggestions of the Reviewers to deal with some additional aspects and/or further develop specific points, this has not been easy to accomplish. The aquaglyceroporins theme has been briefly further exploited.
- Please, provide more information on the lipokine, 12,13-dihydroxy-9Z-octadecenoic acid (page 8, line 392).
As indicated, more information on the lipokine12,13-dihydroxy-9Z-octadecenoic acid (12,13-diHOME) has been provided.
- Please replace “tsunami” by another word (page 9, line 401).
The word “tsunami” has been replaced by “epidemic”.
- Congratulations on Figures design, in particular Figure 2 (page 9).
Thank you, that is very encouraging; we certainly devoted a lot of time to design the figures to better illustrate the content of our manuscript.
- Please integrate the concept of “epigenetics” before the sentence in lines 448-451 (page 10).
As indicated by the Reviewer the concept of “epigenetics” has been integrated before the mentioned sentence.
- Another suggestion from this reviewer is to include the word lifespan at the top of Figure 5 (page 11).
As suggested by the Reviewer, the word “lifespan” has been included at the top of Figure 5.
- Define “critical-window period” (page 11, line 463).
See the Glossary.
- Consider the legend of Figure 5: again, are we being too speculative? I mean: can the potential signature (line 475, page 11) be translated into a scale? An analytical number? Do you think it will be possible to create/develop a scale in the future?
It is expected that artificial intelligence will broadly reshape medicine, thereby improving the experiences of both patients and clinicians. In fact, artificial intelligence is already being applied in an ever increasing number of medical fields moving from what might have been considered speculation years ago to reality right now. Progress in data analysis including image deconvolutions, non-image data sources, unconventional problem formulations, sophisticated algorithms and human–AI collaborations will reduce the gap between research and clinical practice. While these challenges are being addressed, AI’s potential will develop exponentially, making healthcare more accurate, efficient and accessible for patients worldwide (ref Rajpurkar P, Chen E, Banerjee O, Topol EJ. AI in health and medicine. Nat Med. 2022 Jan;28(1):31-38).
- Still regarding Figure 5, how can artificial intelligence-connected information contribute to the development of tools for “exposome” data analysis? (page 11).
As mentioned in the reply to the previous point, AI will be able to relate the huge amount of data of each individual and its surrounding environment to better characterize the specific elements that conform the exposome of the individual.
- Please, reformulate the sentence: “The purported integration proposal will strengthen the better comprehension of the intricate mechanisms underlying the rise of NCDs like obesity, which pose such an important public health problem, and probably would lead to innovative preventive and therapeutic strategies” (page 11, lines 486-489).
As indicated, the sentence has been reformulated.
- Please, provide a comparison between NAFLD, non-alcoholic fatty liver disease and MAFLD, metabolic-associated fatty liver disease supported by adequate bibliographic references (page 12, line 499).
Metabolic dysfunction-associated fatty liver disease (MAFLD) has been proposed to replace the concept of non-alcoholic fatty liver disease (NAFLD) (ref Li S, Xu Z, Li H, Tang J, Liang XY, Tian S, Wu J, Li X, Liu ZL, Xiao J, Chen YL, Wei JY, Ma CY, Wu KN, Ran L, Kong LQ. An Observational and Cross-Sectional Study of the Prevalence of Breast Lesions and Metabolic Dysfunction-Associated Fatty Liver Disease and their Relationship in China. J Gastrointestin Liver Dis. 2022 Mar 19;31(1):31-39). A group of experts have suggested that MAFLD would more accurately reflect not only the disease pathogenesis but would also help in patient stratification for management with NAFLD. However, differences in opinion exist (ref Devi J, Raees A, Butt AS. Redefining non-alcoholic fatty liver disease to metabolic associated fatty liver disease: Is this plausible? World J Hepatol. 2022 Jan 27;14(1):158-167). MAFLD represents a continuum of events characterized by excessive hepatic fat accumulation which can progress to nonalcoholic steatohepatitis (NASH), fibrosis, cirrhosis, and in some severe cases hepatocellular carcinoma. MAFLD might be considered as a multisystem disease that affects not only the liver but involves wider implications, relating to several organs and systems. The MAFLD definition better identifies a group with fatty liver and significant fibrosis evaluated by non-invasive tests (ref Yamamura S, Eslam M, Kawaguchi T, Tsutsumi T, Nakano D, Yoshinaga S, Takahashi H, Anzai K, George J, Torimura T. MAFLD identifies patients with significant hepatic fibrosis better than NAFLD. Liver Int. 2020 Dec;40(12):3018-3030).
- In my opinion, the Conclusions are very well presented (page 12, lines 490-513)
Thank you, that is very encouraging.

Round 2
Reviewer 1 Report
I think that now the work is perfect!
Reviewer 2 Report
It has been revised appropriately to each reviewer's point, respectively.